# Lipids and Fatty Acid Composition Reveal Differences between Durum Wheat Landraces and Modern Cultivars

**DOI:** 10.3390/plants13131817

**Published:** 2024-07-01

**Authors:** Mara Mandrioli, Giovanni Maria Poggi, Giampiero Cai, Claudia Faleri, Marco Maccaferri, Roberto Tuberosa, Iris Aloisi, Tullia Gallina Toschi, Simona Corneti

**Affiliations:** 1Department of Agricultural and Food Sciences (DISTAL), Alma Mater Studiorum, University of Bologna, 40127 Bologna, Italy; mara.mandrioli@unibo.it (M.M.); marco.maccaferri@unibo.it (M.M.); roberto.tuberosa@unibo.it (R.T.); 2Council for Agricultural Research and Economics (CREA), Research Centre for Agriculture and Environment, 40128 Bologna, Italy; giovannimaria.poggi2@unibo.it; 3Department of Life Sciences, University of Siena, 53100 Siena, Italy; giampiero.cai@unisi.it (G.C.); claudia.faleri2@unisi.it (C.F.); 4Department of Biological, Geological and Environmental Sciences (BiGeA), Alma Mater Studiorum, University of Bologna, 40126 Bologna, Italy; simona.corneti2@unibo.it

**Keywords:** lipidomics, wheat kernel, nutritional quality, molecular fingerprinting, genetic variability

## Abstract

Durum wheat (*Triticum turgidum* L. ssp. *durum*) landraces, traditional local varieties representing an intermediate stage in domestication, are gaining attention due to their high genetic variability and performance in challenging environments. While major kernel metabolites have been examined, limited research has been conducted on minor bioactive components like lipids, despite their nutritional benefits. To address this, we analyzed twenty-two tetraploid accessions, comprising modern elite cultivars and landraces, to (i) verify if the selection process for yield-related traits carried out during the Green Revolution has influenced lipid amount and composition; (ii) uncover the extent of lipid compositional variability, giving evidence that lipid fingerprinting effectively identifies evolutionary signatures; and (iii) identify genotypes interesting for breeding programs to improve yield and nutrition. Interestingly, total fat did not correlate with kernel weight, indicating lipid composition as a promising trait for selection. Tri- and di-acylglycerol were the major lipid components along with free fatty acids, and their relative content varied significantly among genotypes. In particular, landraces belonging to *T. turanicum* and *carthlicum* ecotypes differed significantly in total lipid and fatty acid profiles. Our findings provide evidence that landraces can be a genetically relevant source of lipid variability, with potential to be exploited for improving wheat nutritional quality.

## 1. Introduction

Wheat domestication began in the Fertile Crescent around 10,000 years ago, where tetraploid emmer was domesticated from wild emmer. Emmer spread to the Mediterranean basin, where free-threshing tetraploid wheats (*T. turgidum* ssp. *turgidum*) later descended from emmer, and, around 7000 years ago, durum wheats emerged (*T. turgidum* ssp. *durum*) [1,2]. The evolution of tetraploid wheats consists of two domestication steps: primary, from wild to domesticated emmer, and secondary, from emmer to durum wheat [3]. Modern breeding during the Green Revolution led to a new bottleneck in biodiversity as Durum Wheat Landraces (DWL) were progressively replaced by Durum Wheat Cultivars (DWC), specifically selected for traits like semi-dwarf stature, reduced lodging, and higher harvest index [4,5]. Yield-related traits were thus given priority, strongly contributing to the so-called domestication syndrome [6,7,8,9]. Along the centuries, traits associated with nutritional value were also selected, supporting that metabolite profiling can be exploited to study crop domestication [9,10].

This is the case of proteins and carbohydrates, and their distribution in the three main components of the wheat kernel: the embryo, the endosperm, and outer layers. The comparison of DWL and DWC revealed that DWC grain protein content is commonly lower [1,11,12], due to a more developed starchy endosperm.

Although lipids only make up around 2.5–3% of the total kernel, they play a crucial role in the nutritional value and storage stability of flour and wheat-based foods. Lipids interact with gluten proteins and starch, affecting dough functionality [13]. Wheat grains contain a diverse array of lipids, including neutral (acylglycerols and free fatty acids-FFAs) and polar (glycolipids and phospholipids) components. The main storage lipids are triacylglycerols (TAGs), which are found in subcellular organelles called oil bodies. Loaf volume is positively correlated with free polar lipids in flour, but negatively associated with the ratio of free nonpolar to polar lipids, although this is debated [14]. Moreover, the presence of nonpolar lipids is one of the important factors to consider for wheat flour storage, as TAGs can enzymatically be hydrolyzed, yielding diacylglycerols (DAGs), monoacylglycerols (MAGs), and FFAs, finally leading to acid rancidity. Enzymatic oxidative conversion of FFAs can also reduce sensory acceptability [15].

Recently, fatty acid fingerprinting has been identified as an evolutionary signature among tetraploid wheats [9]. Strong divergence emerged between wild emmer and emmer, particularly for the unsaturated fatty acids (UFAs), i.e., mono- and poly-unsaturated fatty acids (MUFAs and PUFAs), significantly reducing the amount of UFAs, mainly gondoic, linoleic, and arachidic acids, which resulted in a decreasing unsaturated/saturated fatty acids ratio (UFA/SFA), impacting kernel health and nutritional quality [16,17]. During domestication, not all metabolic changes necessarily led to an improvement in nutritional quality. Yield-related traits were prioritized over nutritional quality, even though a high ratio of UFA/SFA in the human diet can help prevent cardiovascular diseases [9].

Numerous studies have been carried out to document lipid content and fatty acid composition in whole grains of different wheat species and varieties [9,18]. For example, ancient wheat was found to show higher lipid content than modern wheat, and einkorn was reported to have higher MUFAs, PUFAs, and lower SFAs than durum wheat [19,20]. On the contrary, the bottleneck produced by the Green Revolution, which may have altered lipid composition, has been largely neglected. However, some literature has emerged in recent years, shedding light on this topic. For example, FFAs’ profile could discriminate old vs. modern varieties of both durum and aestivum wheat, using a set of old and modern varieties widely diffused in Central Italy [21]. Most of the significant differences involved four FFAs, i.e., arachidic, oleic, vaccenic, and α-linolenic, sustaining a genetic base for the differences in fatty acid profile between old and modern wheat varieties. Similarly, FFAs’ profile has been proposed to improve authentication and traceability approaches aimed at discrimination of common wheat varieties, including modern and old ones [22].

In this context, the hypothesis of this study is that by employing a lipidomic approach and multivariate data analysis, we can uncover variations in lipid amounts and compositions in the kernels of two durum wheat groups, DWL and DWC. We postulate that the transition from DWL to DWC, which marks the breakpoint of the Green Revolution, may reveal differences in lipid composition. These differences, when identified, could potentially enhance the nutritional values and storage stability of wheat.

## 2. Results

### 2.1. Lipid Content and Lipid Profiles

Free acidity ranged from 3.01 to 5.59 degrees of acidity, with the lowest values observed for Creso and Monastir (3.01 and 3.06 degrees of acidity, respectively) and the highest for Russello, followed by the two DWL-turanicum, i.e., Tetra-ipk814 and Tetra-ipk815 (4.9 and 5.41 degrees of acidity, respectively). Mean free acidity of all samples was 4.3 ± 0.67 degrees of acidity, indicating a good state of preservation. Mean moisture content was 11.24% ± 0.48, with the maximum (12.1%) and minimum (9.7%) values found in Saragolla and AG189, respectively (Table 1). All samples showed comparable humidity, and total fat data were normalized for humidity. The quantitative determination of total fat in flours was evaluated according to the gravimetric total fat method, resulting in a mean fat content of 3.5%.

On average, fat content was significantly higher in DWL compared to DWC, 3.8% vs. 3.2%, respectively (*p* < 0.01) (Figure 1A). However, no significant negative correlation emerged between 1000-seed weight and lipid content, humidity, and free acidity (Figure 1B). Finally, free acidity did not correlate with lipid content or moisture (Figure 1C).

While relative quantification provides only approximate estimations of lipid classes, it is worth noting that among the neutral lipids, TAGs were the most abundant, accounting for 83%, followed by DAGs, FFAs, and MAGs in descending order. The “minor lipids” fraction, comprising hydrocarbons (HCs), tocopherols (TCPs), sterols (STs), and esters (ESs) represented between 4.57 and 7.52% of the total lipids across the fractions studied, having the lowest and highest contents respectively in Kronos and AG189 (4.57% vs. 7.52%).

TAGs negatively and significantly correlated with all the other classes, i.e., DAGs (r = −0.93, *p* < 0.01), FFAs (r = −0.78, *p* < 0.01), and MAGs (r = −0.76, *p* < 0.01) (Figure 2A). DAGs significantly and positively correlated with MAGs (r = 0.78, *p* < 0.01) and FFAs (r = 0.71, *p* < 0.01) (Figure 2B), while MAGs significantly correlated with FFAs (r = 0.84, *p* < 0.01).

As for major lipid classes TAGs, DAGs, and FFAs, no discrimination was found between DWC and DWL using Wilcoxon rank sum test (*p* > 0.05), if DWL sub-groups were not considered; however, significant differences were observed for DWL-carthlicum and DWL-turanicum. The content of TAGs was significantly lower, and the content of DAGs and MAGs was significantly higher, in DWL-carthlicum and DWL-turanicum when compared to DWC and DWL (*p* < 0.05), while their content of FFAs was significantly higher when compared to DWC (*p* < 0.05). Due to their high TAGs and minor component values, Kronos and Kubanka significantly differed from the other genotypes (Figure 3).

PCA was performed to analyze the lipid composition of the genotypes. A total of 72.5% of the total variability was given by differences related to PC1, explained for over 92% by differences in the lipidogram. In general, the prevalence in TAGs corresponded to a reduction in all other classes (FFAs, MAGs, and DAGs). PC2, explained for over 85% by free acidity, was responsible for 16.5% of the overall variability observed. No substantial differences were observed among the genotypes, except for accession belonging to DWL-turanicum, which was characterized by a higher concentration of FFAs, MAGs, and DAGs than the other genotypes. However, DWL showed an enhanced variability in terms of lipidogram, compared to DWC, which instead showed more variability related to PC2 (Figure 4).

From seed cross sections stained with Oil Red O, which shows high affinity for neutral lipids in addition to sterol esters, lipids were mostly localized in the aleurone cells; however, some genotypes contained a significant lipid content also in the starchy endosperm. Cultivars characterized by higher total fat content exhibited more intense staining by Oil Red O, spread not only in the aleurone layer and outermost endosperm, but also diffuse in the innermost endosperm (Figure 5). No difference was highlighted in the embryo, which was in general highly enriched in lipids (Appendix A).

### 2.2. Fatty Acids Profiles

The most abundant FAs were linoleic (C18:2, cis, cis-9,12), oleic (C18:1, cis 9), palmitic (C16:0), α-linolenic (C18:3, all cis-9,12,15), and stearic (C18:0) in decreasing order (Table 2 and Appendix A).

The content of linoleic and oleic acids was discriminating in characterizing individual genotypes. Oleic acid ranged from 11.53 to 17.29 mg/100 mg of fat, with Kubanka and AG189 showing the highest and the lowest oleic acid content respectively. Trinakria and Monastir had the lowest levels of linoleic acid (42.09 and 42.28 mg/100 mg of fat, respectively), while Kyperounda was the most enriched in linoleic acid (46.92 mg/100 mg of fat). In general, DWL-turanicum accessions had the lowest levels of oleic acid while having lower than average levels of linoleic acid (43.56 and 43.35 mg/100 mg of fat for Tetra-ipk814 and Tetra-ipk815, respectively).

Also, the content of palmitic acid (C16:0) varied significantly among accessions. Kronos and Simeto showed a significant reduction in their content (12.34 and 12.35 mg/100 mg of fat respectively), while Tetra-ipk814 and Tetra-ipk815 were significantly enriched (14.64 and 14.50 mg/100 mg of fat, respectively) (*p* < 0.01). Accessions showed a reduced variability for the content of linoleic acid; however, significant differences were found for Trinakria and Monastir, in comparison to Kyperounda, Russello, Kronos, and Haurani (*p* < 0.05). Kyperounda had the highest α-linoleic content, whereas both Odisseo and AG189 had the lowest content (*p* < 0.01).

A deepened comparison among the accessions was performed with cluster analysis for the complete fatty acid profile (complete method), which allowed the accessions belonging to the DWL-turanicum and DWL-carthlicum species to be distinguished from the others, as shown by the cluster dendrogram (Figure 6).

### 2.3. Nutritional Characteristics

The main nutritional parameters, i.e., the ratio ω6/ω3, the PUFA/SFA ratio, and SFA and UFA relative prevalence, as well as AI and TI indices, were calculated for each accession. SFA content differed significantly between DWC and DWL (14.93 and 15.46 mg/100 mg fat, respectively) (*p* < 0.01). On the contrary, no significant differences in PUFA and MUFA content were found between DWC and DWL. Furthermore, significant differences in PUFA/SFA ratio emerged between AG189, Neodur, Kubanka, and AG189 and DWL-*turanicum* accessions (*p* < 0.01). Some DWL had PUFA content greater than 50 mg/100 mg of fat (e.g., Russello, Haurani, and Kyperounda). Among DWL, DWL-turanicum accessions resulted significantly enriched in SFA (*p* < 0.01), with a mean content of 16.58 mg/100 mg of fat. MUFA content was significantly lower in the DWL-turanicum and DWL-carthlicum accessions (*p* < 0.01), with a mean content of 13.08 mg/100 mg of fat. The DWL Kubanka showed the highest amount of MUFA (18.34 mg/100 mg of fat). DWL-turanicum showed a significantly lower UFA/SFA ratio compared to DWC and DWL (*p* < 0.01), due to a reduction in UFA content (60.96 mg/100 mg of fat compared to 63.78 in DWC and 64.04 in DWL) and an increase in SFA content.

The ω6/ω3 ratio was in the range of 9.21 to 16.20; the highest value was obtained for Kyperounda, followed by Neodur and Daurur. These genotypes were significantly different from the others (*p* < 0.01). On the contrary, Odisseo, AG189, and DWL-turanicum were characterized by a significantly lower ω6/ω3 (*p* < 0.01). The TI in the twenty-two accessions ranged between 0.31 and 0.39. No significant differences emerged among accessions, nor between DWC and DWL, with the only exception of DWL-turanicum, characterized by a higher mean TI (0.38) (*p* < 0.01). The mean AI was 0.22; the highest value was obtained for Tetra-ipk814 and Tetra-ipk815 (0.25), which significantly differed from both DWC (mean AI = 0.21) and DWL (mean AI = 0.22) (*p* < 0.01) (Table 3 and Appendix A).

Thus, the distribution of single accessions according to the nutritional parameters revealed interesting differences, as confirmed by the PCA for individuals.

PCA also showed that the increase in SFAs positively correlated with AI and TI and negatively correlated with the PUFA/SFA ratio, while an increase in the content of MUFAs positively correlated with the ω6/ω3 ratio (Figure 7).

## 3. Discussion

Durum wheat flour’s technological properties are mainly affected by starch and protein content. Nonetheless, lipids also significantly impact flour’s technological properties, contributing to its nutritional value and stability [23,24].

Fatty acid fingerprinting has been proposed as a pioneering method for identifying evolutionary signatures in tetraploid wheat [9,16]. However, previous research on lipid content in wheat has focused on domestication steps and elite cultivars [18,25], leaving unexplored changes associated with the selection of yield-related traits during the Green Revolution.

To fill this gap, this study analyzed lipid content and composition in two *T. turgidum* ssp. *durum* groups, representing the breakpoint of the Green Revolution: DWL and DWC. Additionally, two DWL sub-groups, DWL-*carthlicum* and DWL-*turanicum*, were included, representing two sub-groups cropped in restricted geographical areas.

Lipid content was significantly higher in DWL compared to DWC, 3.8% vs. 3.2%, respectively. These were mostly concentrated in the aleurone and cortical endosperm, in accordance with the previous literature [26]. Only accessions characterized by high lipid content showed appreciable lipid amounts in the inner endosperm. Differences in lipid content between DWC and DWL might be partially explicable by the fact that breeding during the Green Revolution favored an increase in kernel size, thus increasing starch and protein content at the expense of lipids. This argument is also supported by comparing DWL average total fat with Svevo, the durum wheat reference variety for transcriptome and quality [27], or by the data for accession AG189, characterized by reduced seed weight and high total lipid content. While no statistically significant negative correlation was observed between 1000-seed weight and total lipid content (*p* > 0.01), a trend of increasing 1000-seed weight as total lipid content decreases was apparent. It is well-established that lipid content is a relatively minor component in wheat, typically ranging from 1.7% in einkorn to 2.9% in spelt [28]. Furthermore, ancient wheat varieties, excluding einkorn, are known to have higher lipid contents compared to common wheat [19,20]. However, it remains necessary to delve deeper into understanding whether this trend holds true across the specific selection steps involved in wheat breeding and development, beyond just the final cultivars. Continued research into the complex interplay between seed size, lipid content, and the evolutionary trajectory of wheat varieties could provide valuable insights into optimizing nutritional profiles while maintaining desirable agronomic traits.

Lipidomic analysis showed a significant correlation between the different classes of neutral lipids. Specifically, TAG negatively and significantly correlated with all other classes. For the major lipid classes, significant differences emerged comparing DWL-*carthlicum* and DWL-*turanicum* with DWC and DWL. These accessions were different, mostly in terms of acidity and lipid composition, both important for kernel and flour shelf-life. Belonging to the same crop year and with the seeds uniformly stored, since the samples show the same moisture degree, we might assume that these are constitutive differences. DWL-*turanicum*, together with DWL Trinakria, showed the lowest content of TAG, and a high amount of FFA compared to the other genotypes, which is an interesting aspect considering the involvement of TAG hydrolysis in the onset of rancidity. The high content in FFA was presumably due to hydrolytic processes, given the low amounts of TAGs and concomitant high amounts of DAGs and MAGs. Given that FFA content is a typical indicator of the degree of hydrolytic rancidity of lipids [10,29], and these accessions showing the highest values for this parameter, it was evident that the DWL-*carthlicum* and DWL-*turanicum* accessions did not undergo a selection process comparable to the other DWL [4,30]. Undoubtedly, these evaluations would greatly benefit from the adoption of a dependable quantification method for multiple lipid classes, as opposed to relying on relative quantification alone. While relative quantification is a widely accepted strategy in nontargeted lipidomics, it only permits rough estimations of lipid classes. Nonetheless, achieving reliable quantification in this field continues to pose significant challenges [31]. In general, free acidity was found to be below the maximum value of six for whole wheat semolina pasta imposed by Italian law [32]. Up to now, these evaluations are a novelty, as previous investigations have been performed only among DWC and near-isogenic lines [33].

Despite high levels of diversity among all accessions, cluster analysis for all fatty acids discriminated indeed only accessions belonging to the DWL-*carthlicum* and DWL-*turanicum* sub-groups. Furthermore, for the most prevalent fatty acids (C16:0, C18:2, C18:3), our results agree with those described in Tavoletti and coworkers [18]. For the two accessions in common, i.e., Svevo and Cappelli, the same fatty acid contents and cultivar-related variations occurred. The analysis properly highlighted constitutive variations among genotypes. Moreover, the high environmental uniformity of the experimental site, and the fact that every trait was evaluated in triplicate, assures consistency to the present work.

As to FFAs’ profile, where mostly arachidic, oleic, vaccenic, and α-linolenic could discriminate old vs. modern varieties of both *T. durum* and *T. aestivum* [21], no significant differences were found in the analyzed germplasm between DWC and DWL, indicating that these traits might have been influenced mostly by the first step of domestication, and not by the following selection steps. However, we found high variability in linoleic and oleic acid content among genotypes, which significantly influenced wheat flour shelf-life. Kubanka had the highest oleic acid content (17.29 mg/100 mg of fat) and an average linoleic acid content (44.64 mg/100 mg of fat), making it particularly interesting from a technological and sustainable perspective. Trinakria demonstrated a good balance between the two fatty acids, with low levels of linoleic acid (42.09 mg/100 mg of fat) and high oleic acid content (14.34 mg/100 mg of fat), making it an interesting genotype for improved shelf-life. In contrast, DWL-*turanicum* accessions had the lowest levels of oleic acid and lower than average levels of linoleic acid (43.56 and 43.35 mg/100 mg of fat for Tetra-ipk814 and Tetra-ipk815, respectively).

The ω6/ω3 ratio in the diet is a commonly cited parameter for evaluating the nutraceutical and nutritional value of foods. The ω6/ω3 PUFA ratio in the whole wheat flour genotypes analyzed ranged from 9.21 to 14.25, which is close to the recommended level of 5:1 to 10:1 [34]. The only exception was Kyperounda at 16.20. However, the absolute intakes of ω3 and ω6 (in grams per day) may be more important for human health than the ratio alone [35].

Numerous studies have demonstrated that individual dietary components, including ω3 PUFAs, can have specific roles in mitigating inflammatory disease states [36,37] by acting as precursors to a range of biologically active compounds, such as prostaglandins and leukotrienes, which regulate immunity, platelet aggregation, and the inflammatory response [38].

However, the benefits of cereal lipids are more complex, as nutrients do not act in isolation but rather have synergistic relationships within the food matrix. Lipids can dissolve bioactive compounds like tocopherols and carotenoids [39,40], and can also enhance the absorption of micronutrients like calcium and iron [41]. Additionally, during food processing, interactions between lipids and other major (i.e., starch and proteins) [42,43,44] and minor (e.g., secondary metabolites, micronutrients) [45,46] components change.

As for nutritional parameters, DWL-*turanicum* accessions showed high AI and TI, and high SFA content (even if not statistically significant, *p* > 0.05). In contrast, DWC showed in general higher PUFA/SFA ratio, and lower AI and TI (but not statistically significant, *p* > 0.05). AG189 showed a low ω6/ω3 ratio (*p* < 0.05). These comparisons are valuable among genotypes, but there should be the awareness that while wheat is a staple food in the Mediterranean basin, it should be integrated into a complete Mediterranean diet, and integrated wheat with other cereals (e.g., barley, millet) and pseudocereals (e.g., quinoa), which on average show lower levels of SFAs, PUFAs, and MUFAs, higher content of UFAs, a lower ω6/ω3 ratio, and consequently lower IA and IT values [47,48,49]. Finally, the diet should be integrated into a Mediterranean lifestyle that incorporates dietary patterns, traditional socio-cultural habits, and food-related practices.

## 4. Conclusions and Future Perspectives

This study demonstrates that the breeding process during the Green Revolution resulted in a reduction in durum wheat grain total lipid content and variability, likely in favor of starch for higher yield and improved pasta-making quality. The data show that the total amount and overall profile of wheat lipids do not negatively affect an important yield trait, 1000-kernel weight, suggesting that lipid composition could be considered to improve cultivar selection, with the goal of obtaining high-yielding genotypes with enhanced nutritional features.

To further validate this hypothesis, analysis of a broader genetic diversity, including landraces, will be crucial. The unique fatty acid profiles and nutritional qualities observed in niche landraces, such as the *turanicum* and *carthlicum* subspecies, represent a valid genetic resource to be explored. Together with the existing literature, this work provides evidence that lipid fingerprinting is an effective method for identifying evolutionary signatures in tetraploid wheat.

Beyond the potential utility of this knowledge for improving genetic resources, there are still important aspects that warrant further investigation. Specifically, it would be valuable to study how the different lipid profiles can impact the conservation of the seeds, as well as the processability and shelf-life of the final wheat products. Deepening these aspects, despite the challenges posed by the complex synergistic interactions involved, could yield insights valuable for technological and nutraceutical applications, as well as for establishing appropriate quality criteria for whole-grain wheat flour products.

Finally, the findings of this research could also inform wheat breeding programs, leveraging landrace genetic resources to transfer desirable quality traits to modern cultivars and enriching nutrient databases like the USDA National Nutrient Database for Standard Reference.

## 5. Materials and Methods

### 5.1. Plant Material and Field Management

Twenty-two durum wheat accessions from the UNIBO-Durum Panel were selected, representing the genetic diversity present in durum wheat (Table 4) [50]. These included thirteen DWCs and nine DWLs, among which two members of *Triticum turgidum* ssp *turanicum* and one member of *Triticum turgidum* spp. *carthlicum*. The first is commonly referred to as “oriental wheat” [51], and has been cropped only on a limited area, comprising Near and Central Asia and Northern Africa [4]. The second is commonly referred to as “Persian wheat”, endemic in Georgia, characterized by resistance to pests [30].

Seed grains were produced in field trial, conducted in 2019/2020 cropping season, at the Agricultural Research Stations of the University of Bologna, Italy (44° 33′ 03″ N, 11° 24′ 36″ E). The site is characterized by a sub-humid climate with an average annual air temperature equal to 12.8 °C and a mean annual precipitation amount of 924 mm. The site is characterized by a high level of uniformity, verified by assessing yield in control plots of cv. Svevo, distributed across the randomized complete block design, thereby minimizing the impact of environmental factors. Experimental layout was a randomized complete block design with three replications. Plot size was 1.5 m × 0.5 m, with a density of 22 plants/m^2^. Fertilizer rate was 112 kg N ha^−1^ and 56 kg P_2_O_5_ ha^−1^, applied at planting. Disease and insect pest pressure were negligible throughout the crop cycle. The seeds were properly dried to 10 to 12% seed moisture content soon after harvesting and stored in paper bags in a climate chamber at 12 °C and 50% humidity. Each trait was evaluated in triplicate, ensuring the robustness of the resulting data.

### 5.2. Reagents and Solvents

All solvents and reagents used were of analytical purity. Petroleum ether (40–60) CAS No. 64742-49-0, n-hexane CAS No.110-54-3, methanol CAS No.67-56-1, ethanol CAS No.64-17-5, sodium hydroxide CAS No.1310-73-2, hydrochloric acid CAS No.7647-01-0, diethyl ether CAS No.60-29-7, water CAS No.7732-18-5, anhydrous sodium sulfate CAS No.7757-82-6, and potassium hydroxide CAS No.1310-58-3 were purchased from Merck (Darmstadt, Germany). Reference standards used are tripalmitin CAS No.555-44-2, tristearin CAS No.555-43-1, dipalmitin CAS No.502-52-3, distearin CAS No.504-40-5, monostearin CAS No.123-94-4, monomyristin CAS No.589-68-4, cholesteryl palmitate CAS No.601-34-3, 5-α-cholestane CAS No.481-21-0, β-sitosterol CAS No.83-46-5, stearic acid CAS No.57-11-4, tridecanoic acid CAS No.638-53-9 were purchased from Sigma-Aldrich (St. Louis, MO, USA). The standard mixture of fatty acid methyl esters (GLC 463) was purchased from Nu-Chek (Elysian, MN, USA). The methyl tridecanoate (>97%) CAS No.1731-88-0 and Oil Red O CAS No. 1320-06-5 were purchased from Sigma-Aldrich (St. Louis, MO, USA), while diazomethane CAS No.334-88-3 in diethyl ether was synthesized in our laboratory.

### 5.3. Seed Staining

The seeds of varieties under examination were transversally cut with a razor blade and lipids were stained with Oil Red O [52]. Oil Red O stock solution (0.5% *w*/*v*) in isopropanol was diluted in distilled water 1.7 times to obtain the Oil Red O working solution. The seeds of varieties under examination were transversally cut with a razor blade and stained with the Oil Red O working solution for 10 min, and then rinsed three times in 70% ethanol. Images were acquired using a stereo microscope (Zeiss Italia, Milan, Italy) under the same light and capture conditions.

### 5.4. Fat Extraction

A Soxhlet glass apparatus was used for the extraction of fat from the analyzed samples according to the reference method [53]. Approximately 10 g of homogenized sample was weighed into an extraction thimble and extracted by petroleum ether for 8 h. Extraction by Soxhlet was performed in triplicate (n = 3) to produce a representative fat for use in subsequent lipidogram and fatty acid methyl ester (FAME) analysis.

### 5.5. Determination of Total Fatty Substances

The determination of the total fatty substances was carried out by applying a gravimetric method as indicated in the Italian official methodology [54]. Briefly, 2 ± 0.0001 g of ground sample was subjected to acid hydrolysis by adding hydrochloric acid for 30 min at 70–80 °C. The total fatty substances were then extracted with equal volumes of diethylethere solvent and petroleum ether at 40–60 °C, to then be quantitatively determined after evaporation of the solvent in a rotavapor and reaching the constant weight in an oven at 100 °C. Results are expressed as a percentage of dry matter (% dry matter). Analysis of each sample was performed three times (n = 3). Results are expressed as a percentage of dry matter. Analysis of each sample was performed three times.

### 5.6. Determination of Main Lipid Classes

The qualitative and semi-quantitative profile of the main lipid classes were determined by Gas Chromatography Shimadzu GC-2010 Plus equipped with an autosampler AOC-20s and a Flame Ionization Detection (GC-FID) (Shimadzu Corp., Kyoto, Japan) according to bibliography [55]. Twenty milligrams of the lipid extract was dissolved in 1 mL of n-hexane, and a quantity equal to 0.88 mg of 5-alpha cholestane standard was added to have a reference on the chromatographic trace, and 1 μL was injected in the GC−FID. A fused silica capillary column (Restek RTX-5, 10 m × 0.25 mm i.d. × 0.1 μm film thickness; Bellefonte, PA, USA), coated with 95% dimethyl- and 5% diphenylpolysiloxane, was used. The temperature was programmed from 100 to 350 °C at 6 °C/min. The injector and FID temperatures were set at 350 °C. The injection was performed in the split mode (1:50), and helium at a flow of 1 mL/min was used as the carrier gas. The calculation was based on an internal percentage calculation of the areas of the various lipid classes identified with the help of reference standards (external standards) used to identify the elution ranges of the major lipid classes. In particular, various mixtures of standards such as monomyristin, monostearin, dipalmitin, distearin, tristearin, trimipalmitin, beta-sitosterol, 5-alpha cholestane cholesteryl palmitate, stearic acid, tridecanoic acid were injected (see Appendix A). Typical profiles of the distribution of lipid classes are reported in Appendix A. Analysis of each sample was performed three times (n = 3). The quantification was obtained by calculating the relative percentage of the areas.

### 5.7. Determination of Total Fatty Acids

European official methods of analysis were used for the qualitative and quantitative determination of the total fatty acids [56,57]. Briefly, 25 mg ± 0.0001 of fat was methylated with diazomethane in diethyl ether, then subsequently 1 mg of internal standard methyltridecanoate was added, and hexane was added up to a total volume of 0.5 mL. Transmethylation was performed by adding 50 µL of 2N methanolic KOH, after 5 min of the reaction the sample was centrifuged and supernatant was injected into GC-FID (Shimadzu GC 2010 PLUS, Kyoto, Japan) under the following conditions: helium carrier gas with a constant flow of 1.08 mL/min, RT-2560 column (biscianopropyl-polysiloxane) 100 m × id 0.25 mm film thickness 0.2-micron Restek with a programmed initial temperature of 100 °C for 1 min, then increase of 3 °C/min up to 180 °C for 10 min then 3 °C/min up to 240 °C maintained for 20 min. Detector temperature 250 °C, injector temperature 250 °C, automatic injection 1 μL split/splitless with split ratio 1:25. Reference standards 463 Nu-Chek Prep Inc were used. Chromatograms were processed with the Shimadzu Software LabSolution version 5.93 GC-Solution 2000–2004. Three independent replicates were performed, and the results are expressed as mg per 100 mg of fat. Representative fatty acid chromatogram is shown in Appendix A.

### 5.8. Determination of Free Acidity

Flour-free acidity was determined in triplicate according to the analytical method of the Italian Istituto Superiore di Sanità, as modified by Acquistucci [58]. Briefly, 4 ± 0.0001 g of ground sample was placed in a glass flask and added with 100 mL of ethyl alcohol at 50 °C. After constant stirring for 3 h, the solution was filtered on a Schull 589/2 paper filter and 50 mL of filtrate was taken and titrated with a sodium hydroxide solution using phenolphthalein as an indicator. Results calculated with the formula below are expressed as acidity degrees corresponding to mL of 1 N NaOH needed to titrate 100 g of flour on a dry matter basis.
$$A=\left[\frac{(\left(V_{1}-V_{2}\right)*c)*2}{m}\right]*\frac{100}{\left(100-U\right)}*\;100$$
where A = Degrees of acidity; V_1_ = mL of sodium hydroxide required to titrate the sample; V_0_ = mL of sodium hydroxide required to titrate the blank; c = Concentration of sodium hydroxide solution; m = Weight of the sample expressed in grams; U = % humidity of the sample.

### 5.9. Lipid Quality Indices

Atherogenicity (AI) and thrombogenicity (TI) indices [59] were calculated according to the formulas reported below:$$AI=\frac{C12:0+4*C14:0+C16:0}{\sum MUFA\;+\sum PUFA\left(n-6\right)+\sum PUFA(n-3)}$$
$$TI=\frac{C14:0+C16:0+C18:0}{0.5*\;\sum MUFA\;+0.5*\sum PUFA\left(n-6\right)+3*\sum PUFA\left(n-3\right)+\frac{\sum PUFA(n-3)}{\sum PUFA(n-6)}}$$

### 5.10. Statistical Analysis

Statistical analysis was performed in R [60]. To test the significance of the diversity among the twenty-two accessions for total lipid profile, FA profile, and main nutritional parameters, One-Way ANOVA with Duncan post hoc test was used. To test the significance of the diversity between DWC and DWL, as well as to test the diversity among DWC, DWL, DWL-*carthlicum,* and DWL-*turanicum*, total lipid content, total lipid profile, and main nutritional parameters were tested using Wilcoxon rank sum test. Correlation analysis (using Pearson’s r) was performed to test the correlation between 1000-seed weight and lipid content, as well as the correlation among the different classes of neutral lipids. Cluster analysis was performed for fatty acid profile (complete method). The proper number of clusters was determined according to the elbow method. PCA was performed by combining lipidogram and free acidity, as well as combining the main nutritional parameters.

## Figures and Tables

**Figure 1 plants-13-01817-f001:**
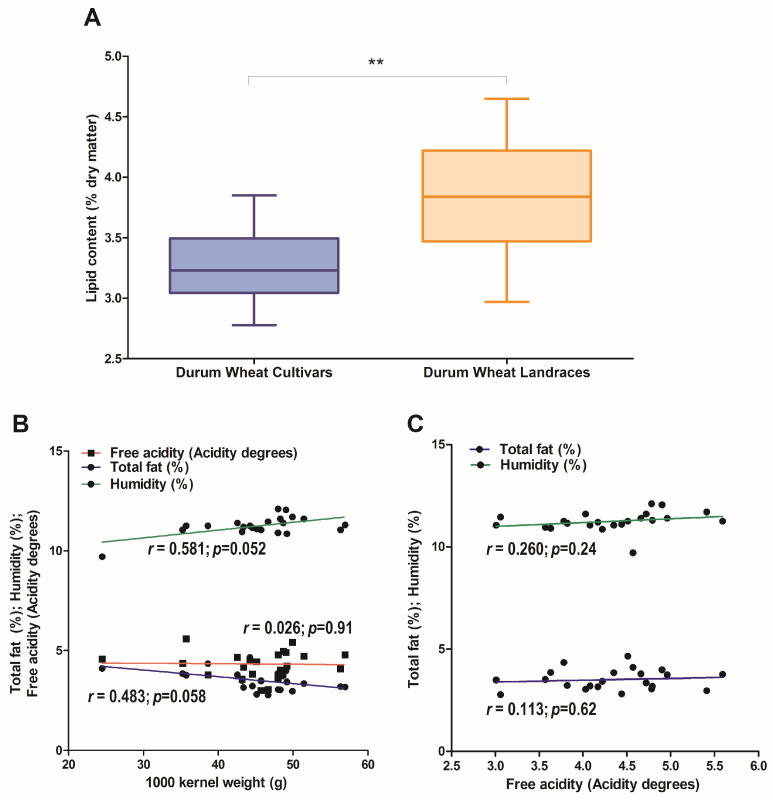
Total lipid content (% dry matter) in the whole meal of the analyzed accessions. Dots represent single accessions, clustered in Durum Wheat Cultivars (DWC) and Durum Wheat Landraces (DWL). Boxes report median (straight lines) and mean (dotted lines) values. Results are mean values for each sample analyzed in triplicate: DWC and DWL were significantly different. ** = *p* ≤ 0.01 (**A**). Correlation analysis between 1000 kernel weight (g) and total fat (%), humidity (%), and free acidity (acidity degrees) (**B**). Correlation analysis between free acidity (acidity degrees) and total fat (%) and humidity (%) (**C**). r correlation values and *p*-values are reported.

**Figure 2 plants-13-01817-f002:**
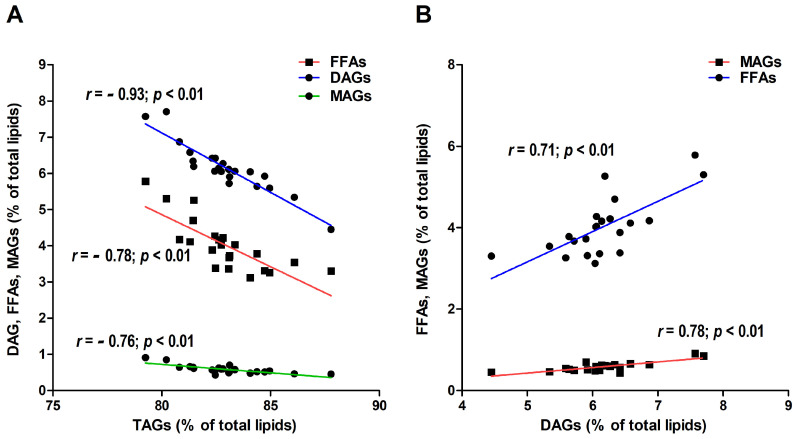
Correlation analysis between triacylglycerol (TAG), free fatty acids (FFA), diacylglycerol (DAG), and monoacylglycerol (MAG) (% of total lipids (**A**). Correlation analysis between DAG and FFA and MAG (% of total lipids) (**B**). r correlation values and *p*-values are reported.

**Figure 3 plants-13-01817-f003:**
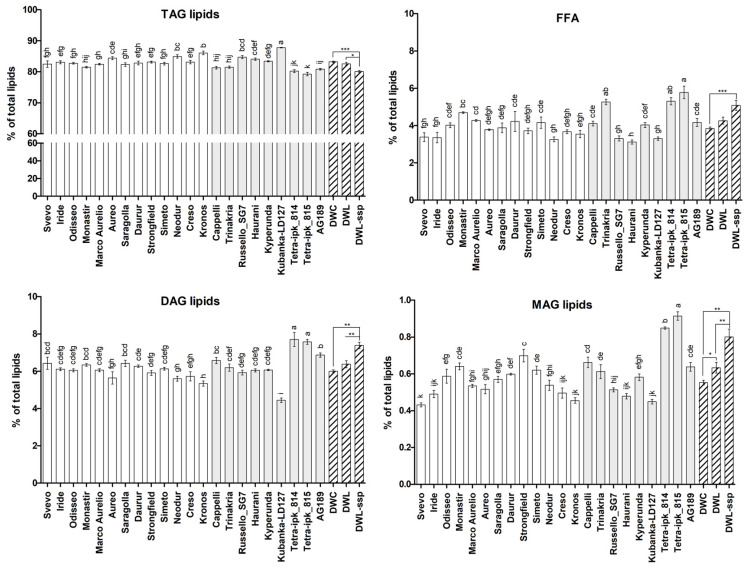
Total lipid profile (% of total lipids) in flours of different wheat genotypes. Major classes of lipids are represented, including triacylglycerol (TAG), free fatty acids (FFA), diacylglycerol (DAG), and monoacylglycerol (MAG). Single DWC accessions are depicted with white bars; single DWL accessions are depicted with grey bars. Bars with the same letter are not significantly different. Mean values of the groups (DWL, DWC, and DWL-ssp) are also reported. Means, reported as bars with oblique lines, marked with asterisks are significantly different: * = *p* ≤ 0.05; ** = *p* ≤ 0.01; *** = *p* ≤ 0.001.

**Figure 4 plants-13-01817-f004:**
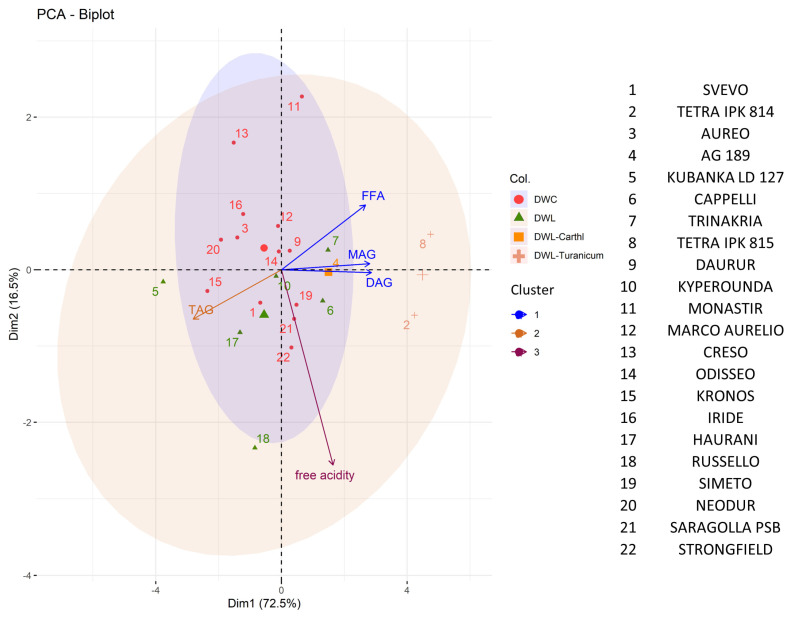
Principal Component Analysis (PCA) of the lipidogram (TAGs, FFAs, MAGs, and DAGs) and free acidity of the wheat genotypes. Axes represent the principal components: Dim1, explaining 72.5% of the variance, and Dim2, explaining 16.5% of the variance. Colors in the PCA indicate variables clustering together.

**Figure 5 plants-13-01817-f005:**
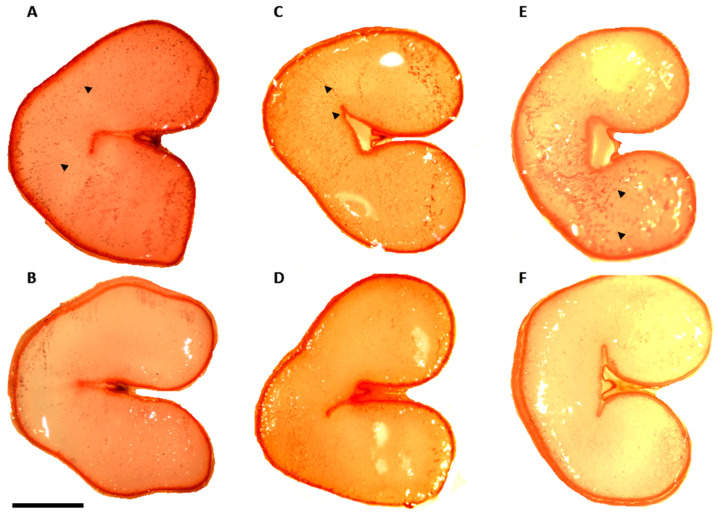
Cross sections of wheat caryopses stained with Oil Red O. Representative images of the three wheat categories analyzed are shown. Durum wheat cultivar: (**A**) Neodur; (**B**) Monastir; durum wheat landrace: (**C**) Haurani; (**D**) Cappelli; durum wheat landrace spp. *turanicum*: (**E**) Tetra-ipk814; (**F**) Tetra-ipk815. Arrows highlight particularly intense and diffuse colorations in the inner endosperm layers. Scale bar = 1 mm.

**Figure 6 plants-13-01817-f006:**
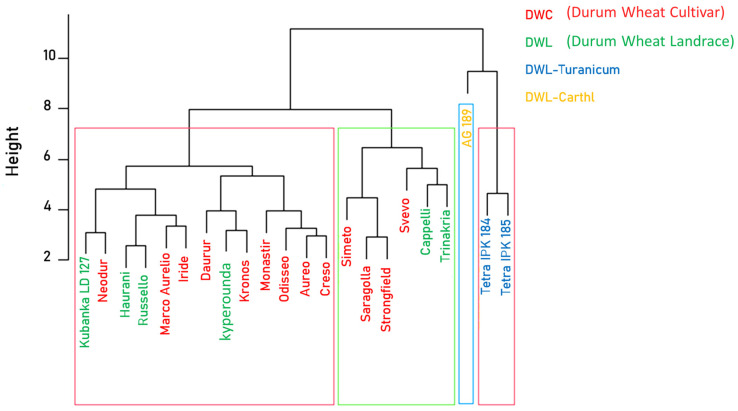
Cluster dendrogram for fatty acid profile (complete method). Font color identifies the wheat category: Durum Wheat cultivar (DWC) in red; Durum Wheat cultivar (DWL) in green; for subspecies, blue highlights DWL-*turanicum*; and yellow DWL-*carthlicum*. Accessions grouped in the same cluster box do not differ in the complete fatty acid profile.

**Figure 7 plants-13-01817-f007:**
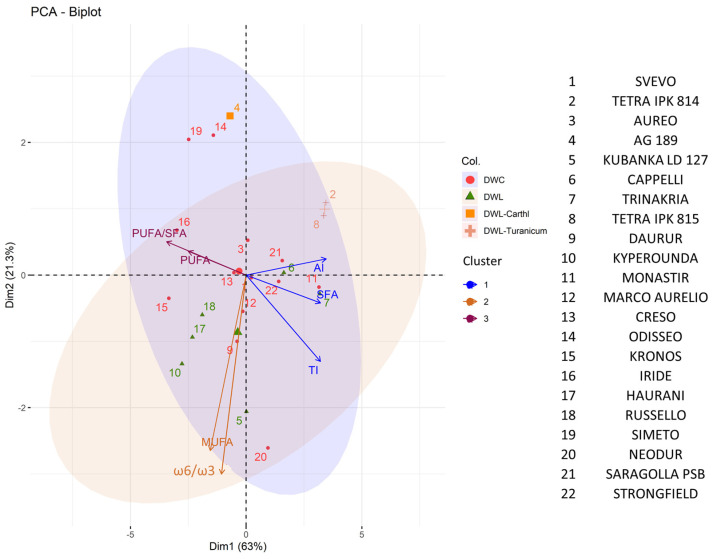
Principal Component Analysis (PCA) of the main nutritional parameters of the twenty-two wheat genotypes. Axes represent the principal components: Dim1, explaining 63% of the variance, and Dim2, explaining 21.3% of the variance. Variables clustering together are marked with the same color.

**Table 1 plants-13-01817-t001:** Weights, total fat (% dry matter), free acidity (acidity degrees), and humidity (%) contents kernel of twenty-two wheat genotypes. Different letters indicate a significant difference to Tukey’s post hoc test. Mean ± standard deviation for Durum Wheat Cultivars (DWC) and Durum Wheat Landraces (DWL) is also reported.

	1000 Kernel Weight (g)	Total Fat (%)	Free Acidity (Acidity Degrees)	Humidity (%)
**Svevo**	45.14 ^b–d^	2.81 ^a^	4.44 ^e–i^	11.1
**Iride**	43.18 ^b–d^	3.51 ^b–f^	3.57 ^a–c^	10.95
**Odisseo**	43.39 ^b–d^	3.16 ^a–c^	4.17 ^c–h^	11.2
**Monastir**	46.67 ^b–d^	2.78 ^a^	3.06 ^a,b^	11.45
**Marco Aurelio**	48.31 ^b–d^	3.04 ^a,b^	4.03 ^c–g^	11.6
**Aureo**	44.58 ^b–d^	3.23 ^a–d^	3.82 ^b–f^	11.15
**Saragolla**	48.00 ^b–d^	3.05 ^a,b^	4.78 ^g–j^	12.1
**Daurur**	49.18 ^b–d^	3.43 ^a–e^	4.22 ^c–i^	10.85
**Strongfield**	48.68 ^b–d^	3.73 ^c–g^	4.96 ^i–k^	11.4
**Simeto**	51.44 ^b–d^	3.35 ^a–e^	4.72 ^g–j^	11.6
**Neodur**	47.98 ^b–d^	3.85 ^d–g^	3.63 ^a–d^	10.9
**Creso**	45.73 ^b–d^	3.48 ^b–f^	3.01 ^a^	11.05
**Kronos**	56.36 ^c,d^	3.20 ^a–d^	4.08 ^c–g^	11.05
**DWC**	**47.59 ± 3.57**	**3.28 ± 0.32**	**4.04 ± 0.61**	**11.26 ± 0.36**
**Senatore Cappelli**	56.95 ^d^	3.33 ^a–e^	4.79 ^g–j^	11.3
**Trinakria**	42.56 ^a–d^	3.78 ^c–g^	4.66 ^g–j^	11.4
**Russello**	35.70 ^a,b^	3.76 ^c–g^	5.59 ^k^	11.25
**Haurani**	44.24 ^b–d^	4.65 ^h^	4.51 ^e–i^	11.25
**Kyperounda**	35.23 ^a,b^	3.84 ^d–g^	4.35 ^d–i^	11.05
**Kubanka**	38.62 ^a–c^	4.34 ^gh^	3.78 ^b–e^	11.25
**Tetra-ipk814**	49.92 ^b–d^	2.97 ^a,b^	5.41 ^j,k^	11.7
**Tetra-ipk815**	49.06 ^b–d^	3.98 ^e–g^	4.90 ^h–k^	12.05
**AG189**	24.49 ^a^	4.10 ^f–h^	4.57 ^f–i^	9.7
**DWL**	**41.86 ± 9.65**	**3.84 ± 0.52**	**4.73 ± 0.54**	**11.22 ± 0.64**

**Table 2 plants-13-01817-t002:** Main fatty acids composition and concentration (mg/100 mg of fat) in the kernel of the twenty-two wheat genotypes. Three technical replicate samples of each genotype were analyzed. Maximum standard deviation (SDmax) for each fatty acid is also reported. Different letters indicate a significant difference based on Duncan’s post hoc test.

	C12:0	C14:0	C15:0	C16:0	ƩC16:1t	C16:1	C17:0	C18:0	ƩC18:1	ƩC18:2t	C18:2	C20:0	ƩC20:1	C18:3	C20:2	C22:0	C22:1	C24:0	C24:1	C22:6
SD_max_	0.01	0.03	0.01	0.5	0.01	0.03	0.01	0.06	0.72	0.02	3.5	0.02	0.05	0.29	0.01	0.02	0.01	0.02	0.01	0.01
Svevo	0.02 ^b,c^	0.09 ^a–c^	0.10 ^e,f^	13.56 ^b–f^	0.10 ^a^	0.16 ^c,d^	0.07 ^b^	1.47 ^a^	14.98 ^c–e^	0.13 ^c–e^	44.93 ^a–d^	0.15 ^b–e^	0.66 ^e,f^	4.08 ^d–f^	0.08 ^g,h^	0.13 ^c–g^	0.08 ^f–j^	0.16 ^f–k^	0.06 ^h,i^	0.04 ^e–h^
Iride	0.01 ^b,c^	0.06 ^c–e^	0.10 ^d,e^	12.53 ^g,h^	0.08 ^d–i^	0.17 ^a,b^	0.06 ^e,f^	0.89 ^c–e^	15.38 ^b–d^	0.10 ^g^	45.58 ^a–c^	0.12 ^h–k^	0.69 ^c–e^	4.17 ^c–e^	0.09 ^e–h^	0.14 ^c–e^	0.08 ^f–j^	0.17 ^e–i^	0.07 ^e–h^	0.04 ^g–i^
Odisseo	0.01 ^b,c^	0.06 ^d,e^	0.09 ^f,g^	12.44 ^gh^	0.08 ^c–g^	0.14 ^e–h^	0.06 ^d–f^	1.00 ^b–e^	13.29 ^h–j^	0.12 ^c–g^	43.28 ^c–e^	0.12 ^i–k^	0.59 ^g^	4.67 ^a,b^	0.09 ^c–h^	0.14 ^c,d^	0.07 ^k^	0.17 ^e–i^	0.06 ^h,i^	0.04 ^g–i^
Monastir	0.01 ^b,c^	0.07 ^c–e^	0.09 ^e–g^	14.41 ^ab^	0.07 ^e–k^	0.14 ^g,h^	0.06 ^e,f^	1.03 ^b–d^	13.75 ^g–j^	0.11 ^e–g^	42.28 ^d,e^	0.11 ^k^	0.67 ^d,e^	3.81 ^f–h^	0.09 ^d–h^	0.12 ^f,g^	0.08 ^g–j^	0.16 ^g–k^	0.07 ^c–f^	0.04 ^h,i^
Marco A.	0.01 ^b,c^	0.06 ^d,e^	0.09 ^e–g^	13.14 ^e–h^	0.07 ^d–j^	0.16 ^c–e^	0.06 ^g^	0.82 ^de^	14.16 ^e–h^	0.11 ^f,g^	43.86 ^b–e^	0.12 ^j,k^	0.71 ^c,d^	3.28 ^j^	0.10 ^c–f^	0.12 ^e–g^	0.09 ^b,c^	0.14 ^j,k^	0.07 ^f–h^	0.05 ^e–h^
Aureo	0.01 ^b,c^	0.06 ^e^	0.09 ^g,h^	13.02 ^f–h^	0.07 ^h–k^	0.16 ^c–f^	0.06 ^e,f^	1.02 ^b–d^	14.22 ^e–g^	0.11 ^e–g^	42.77 ^c–e^	0.14 ^c–g^	0.73 ^c^	4.00 ^d–g^	0.09 ^c–h^	0.15 ^c,d^	0.10 ^b,c^	0.17 ^e–i^	0.07 ^f–h^	0.04 ^e–h^
Saragolla	0.01 ^b,c^	0.07 ^c–e^	0.10 ^e,f^	14.03 ^a–d^	0.07 ^f–k^	0.16 ^b,c^	0.07 ^b–d^	1.08 ^b–d^	13.96 ^f–h^	0.13 ^c–f^	43.75 ^c–e^	0.13 ^e–i^	0.62 ^f,g^	3.98 ^d–g^	0.10 ^b–e^	0.14 ^c–e^	0.08 ^h–k^	0.20 ^c,d^	0.07 ^d–f^	0.06 ^c,d^
Daurur	0.01 ^b,c^	0.06 ^e^	0.08 ^h,i^	12.54 ^g,h^	0.07 ^f–k^	0.11 ^i^	0.06 ^f,g^	1.01 ^b–d^	14.35 ^e–g^	0.12 ^c–g^	43.11 ^c–e^	0.13 ^f–j^	0.65 ^e,f^	2.98 ^k^	0.09 ^e–h^	0.13 ^d–g^	0.08 ^i,k^	0.15 ^h–k^	0.06 ^i^	0.05 ^d–f^
Strongfield	0.01 ^b,c^	0.07 ^c–e^	0.09 ^g,h^	14.16 ^a–d^	0.08 ^b–e^	0.15 ^d–g^	0.07 ^b^	1.13 ^b,c^	13.05 ^i,j^	0.12 ^d–g^	45.52 ^a–c^	0.15 ^b–d^	0.69 ^c–e^	3.59 ^h–j^	0.10 ^b–e^	0.15 ^c^	0.08 ^d–g^	0.18 ^d–g^	0.06 ^h,i^	0.05 ^c–e^
Simeto	0.01 ^b,c^	0.07 ^c–e^	0.11 ^b,c^	12.35 ^h^	0.08 ^d–h^	0.17 ^b,c^	0.07 ^d–f^	0.95 ^c–e^	13.00 ^j^	0.13 ^c–f^	45.24 ^a–c^	0.14 ^d–i^	0.68 ^d,e^	4.44 ^b,c^	0.12 ^a,b^	0.15 ^c,d^	0.09 ^b–d^	0.19 ^c–e^	0.08 ^c–e^	0.04 ^f–i^
Neodur	0.01 ^b,c^	0.07 ^c–e^	0.08 ^h,i^	14.26 ^a–c^	0.06 ^k,l^	0.15 ^d–g^	0.06 ^d–f^	0.99 ^b–e^	16.07 ^b^	0.11 ^f,g^	45.15 ^a–c^	0.13 ^g–k^	0.70 ^c–e^	2.96 ^k^	0.09 ^d–h^	0.13 ^c–g^	0.08 ^f–i^	0.16 ^f–j^	0.07 ^f–h^	0.03 ^i^
Creso	0.01 ^c^	0.07 ^c–e^	0.10 ^d,e^	13.08 ^e–h^	0.07 ^g–k^	0.16 ^c,d^	0.07 ^b–e^	1.13 ^b,c^	14.80 ^d–f^	0.12 ^e–g^	44.40 ^a–e^	0.14 ^c–h^	0.58 ^g^	3.80 ^f–h^	0.08 ^f–h^	0.14 ^c,d^	0.06 ^l^	0.18 ^d–g^	0.06 ^i^	0.05 ^e–g^
Kronos	0.01 ^b,c^	0.05 ^e^	0.08 ^i^	12.34 ^h^	0.07 ^i–k^	0.13 ^h^	0.06 ^e–g^	1.08 ^b–d^	15.46 ^b–d^	0.11 ^f,g^	46.60 ^a,b^	0.14 ^c–f^	0.78 ^b^	3.53 ^h–j^	0.09 ^d–h^	0.13 ^c–f^	0.09 ^b,c^	0.17 ^d–h^	0.08 ^c–f^	0.04 ^h,i^
Cappelli	0.03 ^a^	0.09 ^a–c^	0.11 ^c,d^	13.86 ^a–f^	0.08 ^c–g^	0.18 ^a^	0.07 ^b,c^	1.12 ^b,c^	13.91 ^f–i^	0.14 ^b,c^	43.51 ^c–e^	0.13 ^f–j^	0.65 ^e,f^	3.83 ^f–h^	0.08 ^g,h^	0.14 ^c–f^	0.08 ^e–i^	0.19 ^c–f^	0.07 ^d–g^	0.07 ^b^
Trinakria	0.01 ^b,c^	0.07 ^c–e^	0.13 ^a^	14.33 ^a–c^	0.08 ^c–f^	0.16 ^c–f^	0.07 ^c–f^	1.22 ^b^	14.34 ^e–g^	0.13 ^c–g^	42.09 ^e^	0.14 ^c–f^	0.60 ^g^	3.92 ^e–g^	0.08 ^g,h^	0.15 ^c^	0.07 ^j,k^	0.21 ^c^	0.07 ^e–h^	0.06 ^c^
Russello	0.01 ^b,c^	0.07 ^c–e^	0.07 ^i^	13.47 ^c–f^	0.06 ^k,l^	0.18 ^a^	0.06 ^d–f^	0.82 ^d,e^	15.81 ^b,c^	0.12 ^c–g^	46.63 ^a,b^	0.12 ^i–k^	0.73 ^c^	3.69 ^g–i^	0.09 ^c–g^	0.12 ^f,g^	0.09 ^c–e^	0.14 ^k^	0.07 ^e–h^	0.03 ^i^
Haurani	0.01 ^b,c^	0.07 ^c–e^	0.10 ^e,f^	13.06 ^e–h^	0.07 ^j,k^	0.19 ^a^	0.06 ^f,g^	0.91 ^c–e^	15.99 ^b^	0.12 ^c–g^	46.55 ^a,b^	0.13 ^g–k^	0.67 ^d,e^	3.45 ^i,j^	0.11 ^b–d^	0.13 ^c–g^	0.09 ^d–f^	0.15 ^h–k^	0.06 ^g–i^	0.03 ^i^
Kyperounda	0.01 ^c^	0.07 ^c–e^	0.08 ^h,i^	12.51 ^g,h^	0.07 ^h–k^	0.11 ^i^	0.07 ^b–e^	0.92 ^c–e^	14.57 ^d–g^	0.13 ^c–f^	46.92 ^a^	0.13 ^f–j^	0.68 ^d,e^	2.86 ^k^	0.09 ^e–h^	0.13 ^c–g^	0.09 ^b,c^	0.15 ^h–k^	0.08 ^b–d^	0.04 ^f–i^
Kubanka	0.01 ^bc^	0.05 ^e^	0.08 ^h,i^	13.92 ^a–e^	0.06 ^l^	0.18 ^a,b^	0.07 ^b–d^	1.04 ^b–d^	17.29 ^a^	0.12 ^d–g^	44.64 ^a–e^	0.13 ^f–j^	0.65 ^e,f^	3.38 ^i,j^	0.07 ^h^	0.11 ^g^	0.08 ^e–h^	0.15 ^i–k^	0.08 ^b,c^	0.04 ^g–i^
Tetra-ipk814	0.02 ^b^	0.10 ^a,b^	0.12 ^b,c^	14.64 ^a^	0.09 ^b^	0.18 ^a,b^	0.09 ^a^	1.06 ^b–d^	11.83 ^k^	0.16 ^a,b^	43.56 ^c–e^	0.16 ^a,b^	0.73 ^c^	4.28 ^c,d^	0.11 ^b,c^	0.26 ^a^	0.09 ^b,c^	0.29 ^a^	0.08 ^b–d^	0.05 ^d,e^
Tetra-ipk815	0.02 ^b,c^	0.10 ^a^	0.12 ^a,b^	14.50 ^a^	0.08 ^b–d^	0.17 ^b,c^	0.09 ^a^	1.03 ^b–d^	11.87 ^k^	0.17 ^a^	43.35 ^c–e^	0.17 ^a^	0.71 ^cd^	4.03 ^d–f^	0.11 ^b–d^	0.17 ^b^	0.10 ^b^	0.24 ^b^	0.09 ^b^	0.09 ^a^
AG189	0.01 ^b,c^	0.08 ^b–d^	0.09 ^e–g^	13.30 ^d–g^	0.09 ^b,c^	0.14 ^f–h^	0.07 ^b–d^	0.75 ^e^	11.53 ^k^	0.14 ^c,d^	44.84 ^a–e^	0.15 ^b,c^	1.08 ^a^	4.74 ^a^	0.13 ^a^	0.18 ^b^	0.26 ^a^	0.19 ^c–e^	0.14 ^a^	0.04 ^e–h^

**Table 3 plants-13-01817-t003:** Nutritional characteristics of the twenty-two wheat genotypes. Saturated (SFA), monounsaturated (MUFA), and polyunsaturated (PUFA) fatty acids are reported (mg/100 mg of fat). The ω6/ω3, unsaturated/saturated (UFA/SFA), and PUFA/SFA ratios and the atherogenic (AI) and thrombogenic (TI) indices are reported. Different letters indicate a significant difference to Tukey’s post hoc test.

	SFA	MUFA	PUFA	ω6/ω3	PUFA/SFA	UFA/SFA	AI	TI
Svevo	15.75 ^a^	16.03 ^d–g^	49.27 ^a^	10.90 ^c,d^	3.13 ^a,b^	4.17 ^c–g^	0.21 ^b–f^	0.35 ^b–g^
Iride	14.09 ^a^	16.47 ^e–g^	49.97 ^a^	10.85 ^c,d^	3.55 ^b^	4.74 ^jk^	0.19 ^ab^	0.31 ^a^
Odisseo	14.09 ^a^	14.24 ^a–c^	48.19 ^a^	9.21 ^a^	3.42 ^b^	4.45 ^f–k^	0.20 ^a–d^	0.31 ^a^
Monastir	16.07 ^a^	14.78 ^b–d^	46.32 ^a^	11.02 ^c,d^	2.88 ^a,b^	3.82 ^a–c^	0.24 ^h–j^	0.39 ^g^
Marco Aurelio	14.57 ^a^	15.27 ^c–e^	47.39 ^a^	13.23 ^f,g^	3.25 ^b^	4.32 ^d–i^	0.21 ^b–f^	0.35 ^b–g^
Aureo	14.70 ^a^	15.34 ^c–f^	47.01 ^a^	10.60 ^b–d^	3.20 ^a,b^	4.26 ^d–h^	0.21 ^b–f^	0.34 ^a–e^
Saragolla	15.83 ^a^	14.97 ^c–e^	48.03 ^a^	10.85 ^c,d^	3.03 ^a,b^	4.00 ^a–d^	0.23 ^g–i^	0.36 ^d–g^
Daurur	14.16 ^a^	15.31 ^c–f^	46.34 ^a^	14.25 ^g,h^	3.27 ^b^	4.37 ^e–j^	0.21 ^b–f^	0.35 ^b–g^
Strongfield	16.01 ^a^	14.11 ^a–c^	49.38 ^a^	12.54 ^e,f^	3.08 ^a,b^	3.98 ^a–d^	0.23 ^g–i^	0.38 ^e–g^
Simeto	14.02 ^a^	14.09 ^a–c^	49.97 ^a^	10.13 ^a–c^	3.56 ^b^	4.59 ^h–k^	0.20 ^a–d^	0.31 ^a^
Neodur	15.89 ^a^	17.13 ^g,h^	48.33 ^a^	15.13 ^h,i^	3.04 ^a^	4.14 ^b–f^	0.22 ^e–g^	0.38 ^f,g^
Creso	14.91 ^a^	15.72 ^c–g^	48.44 ^a^	11.57 ^d,e^	3.25 ^b^	4.32 ^d–i^	0.21 ^b–f^	0.34 ^a–f^
Kronos	14.06 ^a^	16.60 ^e–g^	50.36 ^a^	13.11 ^f,g^	3.58 ^b^	4.78 ^k^	0.19 ^a^	0.32 ^a,b^
Cappelli	15.73 ^a^	14.98 ^c–e^	47.62 ^a^	11.19 ^c,d^	3.03 ^a,b^	4.00 ^a–e^	0.23 ^g–i^	0.37 ^d–g^
Trinakria	16.32 ^a^	15.32 ^c–f^	46.28 ^a^	10.59 ^b–d^	2.84 ^a,b^	3.79 ^a,b^	0.24 ^h–j^	0.38 ^g^
Russello	14.89 ^a^	16.95 ^f–h^	50.56 ^a^	12.56 ^e,f^	3.40 ^b^	4.55 ^g–k^	0.20 ^a–d^	0.33 ^a–d^
Haurani	14.61 ^a^	17.07 ^g,h^	50.26 ^a^	13.40 ^f,g^	3.44 ^b^	4.63 ^h–k^	0.20 ^a–d^	0.33 ^a–d^
Kyperounda	14.06 ^a^	15.61 ^c–g^	50.04 ^a^	16.20 ^i^	3.56 ^b^	4.69 ^i–k^	0.20 ^a–c^	0.34 ^a–d^
Kubanka	15.56 ^a^	18.34 ^h^	48.26 ^a^	13.07 ^f,g^	3.10 ^a,b^	4.30 ^d–h^	0.21 ^b–f^	0.36 ^c–g^
Tetra-ipk814	16.72 ^a^	13.00 ^a^	48.15 ^a^	10.09 ^a–c^	2.88 ^a,b^	3.68 ^a^	0.25 ^j^	0.38 ^g^
Tetra-ipk815	16.43 ^a^	13.01 ^a^	47.75 ^a^	10.55 ^b–d^	2.91 ^a,b^	3.72 ^a^	0.25 ^j^	0.38 ^g^
AG189	14.82 ^a^	13.23 ^a,b^	49.90 ^a^	9.39 ^a,b^	3.37 ^b^	4.28 ^d–h^	0.22 ^e–g^	0.32 ^a,b^

**Table 4 plants-13-01817-t004:** Accessions analyzed. DWC = durum wheat cultivar; DWL = durum wheat landrace; DWL-TUR = durum wheat landrace spp. *turanicum*; DWL-CAR = durum wheat landrace spp. *carthlicum*.

Category	Accession Name	Country of Origin	Mega-Environment
DWC	Svevo	ITALY	Southern-Europe
DWC	Iride	ITALY	Southern-Europe
DWC	Odisseo	ITALY	Southern-Europe
DWC	Monastir	FRANCE	Western-Europe
DWC	Marco Aurelio	ITALY	Southern-Europe
DWC	Aureo	ITALY	Southern-Europe
DWC	Saragolla	ITALY	Southern-Europe
DWC	Daurur	FRANCE	Western-Europe
DWC	Strongfield	CANADA	Northern-America
DWC	Simeto	ITALY	Southern-Europe
DWC	Neodur	ITALY	Southern-Europe
DWC	Creso	ITALY	Southern-Europe
DWC	Kronos	US	Northern-America
DWL	Cappelli	ITALY	Southern-Europe
DWL	Trinakria	ITALY	Southern-Europe
DWL	Russello_SG7	ITALY	Southern-Europe
DWL	Haurani	SYRIA	Western-Asia
DWL	Kyperounda	Unknown	Unknown
DWL	Kubanka-LD127	KAZAKISTAN	Central-Asia
DWL-TUR	Tetra-ipk814	IRAQ	Western-Asia
DWL-TUR	Tetra-ipk815	RUSSIA	Eastern-Europe
DWL-CAR	AG189	GEORGIA	Western-Asia

## Data Availability

Data are available upon request to the authors.

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
