# Peer review of "Lipids and Fatty Acid Composition Reveal Differences between Durum Wheat Landraces and Modern Cultivars"

_plants, 2024, doi:10.3390/plants13131817_

Round 1
Reviewer 1 Report
Comments and Suggestions for Authors
Plants - 3069302: Lipids and fatty acid composition reveal differences between durum wheat landraces and modern cultivars
Mara Mandrioli and colleagues have undertaken a lipidomic approach and multivariate data analysis to document differences in lipid amounts and composition in the kernels of two durum wheat groups: modern cultivars and landraces. They compared the fat content and identified the major lipid components between the landraces and the modern varieties. The data was interpreted to show that the landraces and exotic germplasm were valuable source for lipid variability with significantly higher total lipid and fatty acid profiles compared to the modern elite cultivars. The manuscript clearly depicts the extent to which the reported research contributes to filling a knowledge gap in the field of increasing genetic variations through improving the nutritional values and storage stability of modern durum wheat genotypes.
Comments / Minor suggested revisions:
Results:
Page 3 of 20, lines 97 -98 and page 4 of 20, lines 114 - 117. It would be more meaningful if the authors include some figures showcasing results from the correlation analysis. For example, the correlation index between 1,000-seed weight and lipid content and/or correlation between the different lipid classes.
Can Table 2 and Table 3 be reorganised for better comprehension? The current format makes it almost impossible to read. The authors can set the text characters representing the various similarities/differences between the different genotypes as a superscript instead of being on the same line as the text values.
Materials and methods
The authors stated that the field trial site had a high level of environmental uniformity, particularly for the yield-based traits was any analysis conducted to support the acclaimed uniformity of data obtained across the three replicates from the experimental design?
Align the words on page 14 lines 346 and 347 and add the missing comma on line 352.
I suggest rearranging the sentences from line 363 – 368 for better comprehension. For example: “Oil Red O [26] stock solution (0.5% w/v) in isopropanol was diluted in distilled water 1.7 times to obtain the Oil Red O working solution. The seeds of varieties under examination were transversally cut with a razor blade and stained with the Oil Red O working solution for 10 mins, and then rinsed three times in 70% ethanol. Images were acquired using a Zeiss stereo microscope under the same light and capture conditions”.
Please correct the error on the Supplementary Table 2 title. On page 2 of 20, line 64, unsaturated fatty acid was represented by the acronym (UFA). However, the Supplementary Table 2 title assigned SFA for unsaturated fatty acid.
To increase accessibility, include the heatmap scale and change the colour scheme for Supplementary Table 1 and 2 so that it is easier for those who are colour blind (i.e. not red/green).
Comments on the Quality of English Language
Comments indicated on the attached file.
Author Response
Dear reviewer 1,
thank you for the time and effort you have put into reviewing our manuscript, which was appreciated, as it denoted accuracy and proactiveness to improve the manuscript. We addressed all the issues and present a reviewed version of the manuscript, which presents the main corrections in red font, in order to better follow the reviewing steps.
The most relevant corrections/modification are listed below:
- We have added new figures to the manuscript that showcase the results from the correlation analysis. Correlation analysis graphs have been included in Figure 1 (see B and C panel) and in a new figure (figure 2). Specifically, we have included figures showing the correlation index between seed parameters (1000 kernel wight, humidity, total fat and free acidity) (Figures 2B and C), as well as the correlation between the different lipid classes (Figure 2).
- We have reorganized all the tables according to your suggestions to improve comprehension. We have set the text characters representing the various similarities/differences between the genotypes as superscripts, making the tables easier to read. We thank you a lot for these suggestions.
Alongside, the accessibility of the heatmaps in supplementary material has been improved and the colour scale changed, respectfully to readers who are colour blind.
- Materials and Methods was improved in order to support the claimed uniformity of the data obtained across the three replicates from the experimental design. Moreover, some parts have been rephrased, according to your suggestions.
- Formatting, typos and language have been corrected according your and reviewers’ suggestions, and after the correction of the text by all the authors. Please, main changes tracked in red font.
- Please note that abstract, introduction and discussion have been partially rewritten or implemented. Some old references have been removed and new background and relative bibliography have been added.
We hope that addressing all the reviewers’ comments and suggestions, the new version of the manuscripts results increased in concepts, clarity and general layout.
Thank you, the authors

Reviewer 2 Report
Comments and Suggestions for Authors
This is important study and revealed that Lipids and fatty acid composition reveal differences between durum wheat landraces and modern cultivars. Article has been arranged in a good order, and all chapters are linked with eachother, also presentation of the results are excellent and after reading of this article i have some major changes with the acceptance of this quality work.
1. Please make it clear in the abstract that what are the points to raise your attentation on these points?
2. Add a valid conclusion in abstract.
3. Introduction is not enough, specially for lipid and fatty acid composition among different tradational vs modern cultivars, you can also add some literature from other cereals crops.
4. Table 1 and other Tables where you have used statistical letters for significancy such as (abcd, cdefghi etc..) must be replace with e.g., (a-d, c-i) for understanding and to make it more clear for the readers.
5. Table 1, what is DWC and DWL, you can explain in the footnote.
6. Add the y and x-asixes of Figure 1.
7. Comment for Table 1 can be also followed for Table 2 and Table 3.
8. I suggest to add sub-heading in the discussion section, and if possible you can add some more literature.
9. I not find conclusion section, please confirm.
Author Response
Dear reviewer 2,
thank you for the time and effort you have put into reviewing our manuscript, which was appreciated, as it denoted accuracy and proactiveness to improve the manuscript. We addressed all the issues and present a reviewed version of the manuscript, which presents the main corrections in red font, to better follow the reviewing steps.
The most relevant corrections/modification are listed below:
- Abstract has been almost entirely rewritten, aiming at being more meaningful both in aims and in conclusions, without exceeding the 200 words suggested by the journal.
- Many efforts have been made to improve text, both in content and in clarity. Some old references have been removed and new background and relative bibliography have been added both in the introduction and in the discussion. We aimed at improving concepts and report on the published bibliography on the topic, according to your suggestion. Parallelly, we tried not to excess too much the length of these two sections, and we hope that the new version of the manuscript results balanced.
- We strongly worked on Figures and Tables also, and improved clarity. First of all, we have reorganized all the tables according to your suggestions to improve comprehension. We have set the text characters representing the various similarities/differences between the genotypes as superscripts, making the tables easier to read. We thank you a lot for these suggestions. Tables were also implemented in accessibility, by explaining abbreviations. To note is that the accessibility of the heatmaps in supplementary material has been improved and the colour scale changed, respectfully to readers who are colour blind (this was a useful suggestion of reviewer 1).
Abbreviations have been avoided also in Figures (as suggested by reviewer 3). See for example Figure 6 (ex Figure 5). The new version of the manuscript results implemented also in Figure 1 (now reporting also correlation analysis) and by the addition of a new Figure (Figure 2), asked by reviewer 1. As the number of figures has been increased, we checked and corrected Figure’s numeration and citation along the text.
- While we are not sure it’s allowed to create sub-headings in the discussion section, this has been strongly implemented and thank to your suggestion, we created a section “conclusion and future perspectives”.
Formatting, typos and language have been corrected according your and reviewers’ suggestions. Please, main changes tracked in red font.We hope that the new version of the manuscripts results increased in concepts, clarity and general layout.
Thank you, the authors

Reviewer 3 Report
Comments and Suggestions for Authors
The article "Lipids and fatty acid composition reveal differences between durum wheat landraces and modern cultivars” by Mandrioli et al. addresses to determine the differences in lipid amounts and compositions between durum wheat landraces and modern cultivars, and to understand the implications of these differences for nutritional and breeding purposes. This study is highly relevant to the field of plant breeding and nutrition science. It addresses a specific gap concerning the lipidomic profiles of durum wheat landraces compared to modern cultivars. While previous research has focused on major kernel metabolites, this paper highlights the importance of minor bioactive components like lipids, which have significant nutritional and nutraceutical benefits.
It reveals that landraces have higher fat content and distinct fatty acid profiles compared to modern varieties. The conclusions of the study are consistent with the evidence and arguments presented. The finding that landraces have higher lipid content and unique fatty acid profiles is well-supported by the data. The interpretation that these differences are a result of selection for yield-related traits during the Green Revolution is logical and aligns with historical breeding objectives. The references cited are appropriate and relevant, encompassing key studies on wheat lipidomics, plant breeding, and nutritional analysis. This study significantly contributes to the understanding of lipid composition in durum wheat, highlighting important differences between landraces and modern cultivars.
Therefore, I recommend publishing the article in Plants after just the following minor revision:
- The caption for tables 1 and 3 must be in text format above the table and not be part of the table;
- In figures 1 and 5, the abbreviations DWC and DWL must be written in full.
Author Response
Dear reviewer 3,
thank you for the time and effort you have put into reviewing our manuscript, which was really appreciated. We addressed all the issues and present a reviewed version of the manuscript, which presents the main corrections in red font, to better follow the reviewing steps.
The most relevant corrections/modification are listed below:
- We strongly worked on Figures and Tables also, and improved clarity. First of all, we have reorganized all the tables according to your suggestions to improve comprehension. We have set the text characters representing the various similarities/differences between the genotypes as superscripts, making the tables easier to read. We thank you a lot for these suggestions. Tables were also implemented in accessibility, by explaining abbreviations. To note is that the accessibility of the heatmaps in supplementary material has been improved and the color scale changed, respectfully to readers who are colour blind. Abbreviations have been avoided also in Figures (as suggested by reviewer 1). See for example Figure 6 (ex Figure 5). Please, have a look at the new version of Figure 1 (now reporting also correlation analysis) and at the new figure 2, asked by reviewer 1. As the number of figures has been increased, we checked and corrected Figure’s numeration and citation along the text.
- As you’ll notice, the abstract has been almost entirely rewritten, aiming at being more meaningful both in aims and in conclusions, according to the request of reviewer 2. The other two reviewers asked extensive implementation in the text, so you’ll find several parts added and/or rewritten and implemented in bibliography. More in detail, some old references have been removed and new background and relative bibliography have been added both in the introduction and in the discussion. We aimed at improving concepts and report on the published bibliography on the topic, according to your suggestion. Parallelly, we tried not to excess too much the length of these two sections, and we hope that the new version of the manuscript results balanced.
- We created a section “conclusion and future perspectives”.
Formatting, typos and language have been corrected according all the reviewers’ suggestions. Please, main changes tracked in red font.
We hope that we have addressed all of your comments and suggestions, and we hope that these revisions have improved the quality and clarity of our work.
Thank you, the authors

Round 2
Reviewer 2 Report
Comments and Suggestions for Authors
This draft has been significantly improved and accepted for publication.
Author Response
Dear Reviewer,
Thank you for approving our paper for publication. We appreciate your feedback.
The authors